# Perception of Pathologists in Poland of Artificial Intelligence and Machine Learning in Medical Diagnosis—A Cross-Sectional Study

**DOI:** 10.3390/jpm13060962

**Published:** 2023-06-07

**Authors:** Alhassan Ali Ahmed, Agnieszka Brychcy, Mohamed Abouzid, Martin Witt, Elżbieta Kaczmarek

**Affiliations:** 1Department of Bioinformatics and Computational Biology, Poznan University of Medical Sciences, 61-806 Poznan, Poland; 2Doctoral School, Poznan University of Medical Sciences, 61-806 Poznan, Poland; 3Department of Clinical Patomorphology, Heliodor Swiecicki Clinical Hospital of the Poznan University of Medical Sciences, 61-806 Poznan, Poland; 4Department of Physical Pharmacy and Pharmacokinetics, Poznan University of Medical Sciences, 60-806 Poznan, Poland; 5Department of Anatomy, Rostock University Medical Centre, 18057 Rostock, Germany; 6Department of Anatomy, Technische Universität Dresden, 01307 Dresden, Germany

**Keywords:** artificial intelligence, machine learning, histopathology, medical diagnosis, cancer, tumor pathology

## Abstract

Background: In the past vicennium, several artificial intelligence (AI) and machine learning (ML) models have been developed to assist in medical diagnosis, decision making, and design of treatment protocols. The number of active pathologists in Poland is low, prolonging tumor patients’ diagnosis and treatment journey. Hence, applying AI and ML may aid in this process. Therefore, our study aims to investigate the knowledge of using AI and ML methods in the clinical field in pathologists in Poland. To our knowledge, no similar study has been conducted. Methods: We conducted a cross-sectional study targeting pathologists in Poland from June to July 2022. The questionnaire included self-reported information on AI or ML knowledge, experience, specialization, personal thoughts, and level of agreement with different aspects of AI and ML in medical diagnosis. Data were analyzed using IBM^®^ SPSS^®^ Statistics v.26, PQStat Software v.1.8.2.238, and RStudio Build 351. Results: Overall, 68 pathologists in Poland participated in our study. Their average age and years of experience were 38.92 ± 8.88 and 12.78 ± 9.48 years, respectively. Approximately 42% used AI or ML methods, which showed a significant difference in the knowledge gap between those who never used it (OR = 17.9, 95% CI = 3.57–89.79, *p* < 0.001). Additionally, users of AI had higher odds of reporting satisfaction with the speed of AI in the medical diagnosis process (OR = 4.66, 95% CI = 1.05–20.78, *p* = 0.043). Finally, significant differences (*p* = 0.003) were observed in determining the liability for legal issues used by AI and ML methods. Conclusion: Most pathologists in this study did not use AI or ML models, highlighting the importance of increasing awareness and educational programs regarding applying AI and ML in medical diagnosis.

## 1. Introduction

### 1.1. The Role of Artificial Intelligence (AI) and Machine Learning (ML) in Medical Diagnosis

Since the early 2000s, artificial intelligence (AI) and its advanced subtypes, machine learning (ML) with deep learning (DL), have gained significant prominence in the medical field. They are tools used to develop diagnostic algorithms, predict a patient’s survival probability, guide medical diagnostic processes, and suggest the proper treatment protocol [1]. The integration of AI and ML into medical diagnosis requires a massive amount of data collection, training, and testing. However, the utilization of big data in this context encounters several barriers [2]; firstly, the sheer size of the data poses challenges regarding storage, management, and processing. Furthermore, big data comprises diverse types of structured, unstructured, and semi-structured data, making integration and analysis complex. Additionally, the rapid generation of data and processing speed can overwhelm traditional systems. Extracting meaningful insights and value from big data requires advanced analytical tools and efficient processing and analytical techniques. Nevertheless, numerous researchers and scientists have successfully overcome these obstacles and developed several effective AI/ML algorithms currently employed in medical diagnosis [3]. The computational approach in pathology is widely recognized and used in scientific, quality assurance, color normalization, and diagnostic processes. These approaches can help with tasks including identifying patterns in tissue samples, quantifying the presence of specific proteins or other molecules, and automating the diagnosis of diseases. They can also be used to improve accuracy and efficiency of diagnostic processes, reducing the possibility of human errors and supporting research into the underlying causes of diseases [4].

Nowadays, pathologists increasingly rely on AI/ML in medical diagnosis. There are several applications of AI with ML in this context. Firstly, in image analysis, AI algorithms can be trained to analyze images of tissue samples and detect patterns or features indicative of specific diseases. This capability proves particularly valuable in identifying early stage or subtle disease signs that may elude visual observation [5]. Secondly, in data analysis, pathologists generate substantial amounts of data during their work, including laboratory results and patient records. AI algorithms can analyze this data and identify patterns or associations that may not be discernible to human analysts [6]. Eventually, AI systems can be used in decision-making to provide pathologists with recommendations or guidance on the most appropriate course of action for a given patient based on the available data and evidence [7]. Overall, integrating AI in pathology can improve the accuracy and efficiency of medical diagnosis and support pathologists in their work.

### 1.2. Artificial Intelligence and Machine Learning Drawbacks and Legal Considerations

It is of utmost importance to underscore the potential drawbacks of AI/ML with the daily practice of pathologists. Firstly, from a legal standpoint, pathologists must assume responsibility for the diagnostic process. Consequently, integrating AI into the daily practice of pathologists may inadvertently prolong the diagnostic timeline, which may not be advantageous in some instances [8]. Furthermore, it is crucial to acknowledge that AI/ML algorithms can be susceptible to biases like their human counterparts. These biases can manifest at various stages of the AI system’s development and deployment, including the data used to train the algorithm, its design, or its practical application. Such biases could have profound implications for the involved patients, potentially resulting in substandard care or the denial of specific treatments. If an AI system fails to identify critical cases, patients may not receive timely or appropriate care, leading to severe or fatal consequences [9]. Moreover, there are considerations of ethical and psychological implications of misdiagnosis for healthy patients. They may undergo unnecessary and potentially harmful treatments and experience anxiety, stress, and other psychological impacts due to misdiagnosis. Additionally, the misdiagnosis may have financial consequences if the patient must pay for unnecessary medical procedures or treatments. The consequences can also be significant, even for sick patients wrongly diagnosed as healthy. They may not receive the appropriate treatment for their condition, which could worsen it or even lead to death. In some cases, the misdiagnosis may also lead to delays in getting the proper treatment, which can seriously harm the patient’s health [10]. It is crucial to address the perspective of oncology specialists regarding this matter. Oncologists heavily rely on pathology examination results to inform their approach when treating cancer patients [11]. Pathologists play a critical role in cancer diagnosis, staging, and treatment response monitoring. Some oncologists hesitate to adopt AI in pathology due to their need for greater familiarity with the technology or concerns regarding its accuracy and reliability. Conversely, other oncologists express enthusiasm for the potential benefits of AI/ML and eagerly seek to incorporate these advancements into their clinical practice [12]. Ultimately, the adoption of AI in pathology will depend on various factors, encompassing the availability of dependable and accurate AI systems, the willingness of pathologists and oncologists to embrace the technology, and the establishment of regulatory and legal frameworks that support its utilization [13].

### 1.3. Highlighting the Gap and Hypothesis Formulation

Cancer is the second cause of death in Poland. The diagnoses will only grow with the gradually aging Polish population [14]. Pathological diagnosis of cancer is a vital part of oncology treatment, and pathology is the bottleneck in the treatment of oncology patients. The universal health care in Poland is being heavily underfinanced, resulting in a low number of active pathologists which is preventing swift and proper diagnosis. In Poland, there are 783 active pathologists. This makes the number of two active pathologists per 100,000 citizens one of the lowest in European countries [15]. It is speculated that those numbers are actually much lower. A more realistic number of active pathologists in Poland oscillates around 425, resulting in one pathologist per 84,000 Polish citizens, while the European average is one pathologist per 40,000 citizens [16]. Data from 24 European countries and the USA and Canada were investigated. Germany has the second-lowest number of pathologists per population in Europe, with only one pathologist for every 47,989 citizens (mean: 32,018) [17]. It is also worth noticing the significant age gap among those medical professionals, as 72% of all active pathologists are above 50 years, and 42% are above 60 years old. This means there will be a significant drop in the number of active pathologists in the near future, significantly impairing the efficiency of oncological treatment. At the moment, there are 230 slots for pathology residency, only 130 of which are occupied. The reasons behind the unpopularity of pathology among young medical doctors are, among others, the overwhelmingly vast knowledge young people need to attain during their training, fewer opportunities for additional earnings, a huge responsibility in the diagnostic process, and the longer time required for a pathologist to become more independent in the process of diagnosis [18]. To the best of our knowledge, no study has focused solely on the situation in Poland. Therefore, our study is designed to measure the attitude of the pathology community in Poland toward AI/ML in medical diagnosis, their knowledge, experiences, concerns, and their compatibility with AI/ML applications in medical diagnosis. We will also investigate the possibility of predicting whether the pathologist used or did not use AI/ML earlier based on their answers.

## 2. Materials and Methods

### 2.1. Study Design

We performed a cross-sectional study using an anonymous, self-administered, and structured online survey tool through the “Microsoft Forms” platform.

### 2.2. Study Population

The inclusion criteria were applied to all individuals who agreed to participate in the study: age ≥ 28 years, live in Poland, and specialized in pathology. There were no restrictions on gender, age, or socioeconomic level. The survey was distributed in June–July 2022 using two mailing lists: (i) an internal mailing list for Poznan University of Medical Sciences working pathologists; and (ii) a mailing list created by the Department of Bioinformatics and Computational Biology in collaboration with the University Clinical Hospital in Poznan. These lists only included some pathologists in Poland, and pathologists were encouraged to share the survey in their departments with only those included in our including criteria.

### 2.3. Study Tool

Questions were created based on previous literature that investigated the use of AI/ML in clinical pathology and healthcare [3,19,20,21]. The survey is composed of two parts: (1) the demographic characteristics (age, gender, medical years of experience, and specialty); (2) previous experience with AI/ML, trust in AI/ML results, and its efficacy in cancer diagnosis. This section also included the level of agreement with AI/ML’s space-time limitation and its validity in controversial cases, and its ability to save money and time, replace human doctor positions, speed up the diagnosis process, reduce medical errors, assist in complex cases, be applied to every patient, produce high-quality data, and sympathize with patients. Finally, we asked the pathologist to determine their point of contact to issue the final decision on the medical diagnosis, which field of medicine the AI/ML could be helpful in, and who should be liable for legal problems caused by AI/ML. They also expressed their thoughts in a short answer about the future of AI/ML in the medical field (the survey is available in the Appendix A). The survey was then pilot tested by four pathologists at University Clinical Hospital in Poznan for face validity and time to complete. We edited the format of the paper and online survey from the pilot test feedback. The scales’ internal consistency reliability was determined with Cronbach’s standardized alpha which quantifies the level of agreement on a scale between 0 and 1. A higher value indicates higher agreement. This also ensured consistency, indicating the measurements are reliable and the items might measure the same characteristic [22].

### 2.4. Statistical Analysis

We performed the statistical analysis using IBM^®^ SPSS^®^ Statistics v.26 and PQStat Software v.1.8.2.238. The Shapiro–Wilk test checked normality, and normally distributed data were reported as mean ± standard deviation (SD), and non-normally distributed data were reported as the median and interquartile range (IQR). The Mann–Whitney U test calculated differences between agreement levels between AL/ML users and non-users. Moreover, the Chi-squared (χ^2^) test measured the significance of frequencies associated with AL/ML concerns. Finally, a stepwise multi-logistic regression model was used to study the predictors of AL/ML users. Logistic regression results were presented as odds ratios (ORs) and 95% confidence interval (95% CI). The computed *p*-value < 0.05 is considered statistically significant for all tests except in Table 3 due to applying Bonferroni correction, where *p*-value less than 0.00384 is considered statistically significant. Text mining on RStudio Build 351 was used to analyze the pathologists’ comments and generate the word frequency map.

## 3. Results

### 3.1. Demographic Data of the Participants

A total of 68 pathologists from Poland participated in the survey study. Cronbach’s standardized alpha demonstrated a good reliability of α = 0.79 [22] (Table 1).

The median age of the participating pathologists was “37 (33–41.5)” years. Approximately 53% of the physicians were females, and 37% were males. The pathologists have a median of “10 (6.75–15.25)” years of medical experience. Regarding previous experiences with AI/ML, 42.19% of the pathologists have answered (Yes) while 57.81% answered (No). Moreover, on a scale of ten, the pathologists expressed their trust in the AI/ML results, with a median of “7 (5.75–8)”. Similarly, they reported the efficacy of AI/ML in diagnosing cancer cells with a median of “7 (6–8)” (Table 2).

### 3.2. Level of Agreement with AI/ML in Clinical Use

The frequency of the agreement with the use of AI and ML is displayed in Figure 1. We found a significant difference (*p* < 0.001) between the self-reported knowledge of pathologists who have used AI/ML before, and those who have never used AI/ML before (100% and 41.67%, respectively, as shown in Table 3, Q1). The remaining results were comparable between those with previous experience and those who have never used AI. Both groups agreed on the value of AI applications in the medical field (Q2), that AI/ML applications would save time and money for physicians (Q4), and that they could speed up the diagnosis process (Q6). Pathologists also agreed that AI might assist in reducing medical errors (Q7), providing high-quality data (Q8), and having no space-time limitations (Q9) as AI shares computational storage and framework with the environment [23]. However, pathologists also agreed that AI might be challenging to apply to controversial subjects (Q12) and cannot empathize with a patient’s emotional well-being (Q13). Additionally, pathologists in this study disagreed that AI has better diagnostic ability (Q3), that it will replace their work in the future (Q5), and that it applies to every patient (Q11), as shown in Table 3. Furthermore, a positive correlation (rSpearman = 0.8, *p* < 0.001) was reported between trust and efficacy of AI/ML models in cancer diagnosis (Figure 2).

In stepwise multi-regression analysis (Table 4), we found that pathologists with previous experience in AI/ML were almost 18 times more likely to claim they know AI/ML (OR = 17.9, 95% CI = 3.57–89.79, *p* < 0.001). Moreover, previous users of AI/ML also had higher odds of reporting satisfaction with the speed of AI/ML in the medical diagnosis process (OR = 4.66, 95% CI = 1.05–20.78, *p* = 0.043).

### 3.3. Concerns with AI/ML

In this section, we asked the pathologists three questions: (I) If your medical diagnosis differs from the AI diagnosis, which will you follow; (II) In which field of medicine do you think AI will be most helpful; and (III) Who do you think will be liable for legal problems caused by AI, (Figure 3). For the first question, the previous users and those who never used AI/ML selected the physician’s opinion with 92.59% and 91.89%, respectively. Similarly, in the second question, both the previous users and those who had never used AI/ML highlighted the benefits of AI/ML in making diagnoses (59.26% vs. 42.86%) and in biomedical research (29.63% vs. 34.29%), respectively. However, the differences between the results of the “users” and the “never used” were not statistically significant in these two questions.

Finally, interestingly we found that in the third question, there was a significant difference (*p* = 0.003) between pathologists who had previous experience in AI/ML and those who never had before. The pathologists with previous experience selected that physicians would be liable for legal problems caused by AI (83.33%). In comparison, the pathologists who had never had experience in AI/ML before selected the AI-developing company (55.56%) (Figure 3).

### 3.4. Future Expectations for AI/ML

Participating pathologists had the opportunity to express their opinions. Most pathologists agreed on the importance of AI/ML in medical diagnosis and how much it would be helpful and save time for them in daily work. Moreover, they believe AI/ML will provide a supportive opinion based on the model’s accuracy during the treatment process and decision-making protocol. However, the majority disagreed that AI/ML will replace their position in the near future. Instead, they think it will be an assisting tool. Finally, the pathologists wish that the government could provide more resources and training related to AI/ML in medical diagnosis generally and in different areas of specialization (Figure 4).

## 4. Discussion

This study was conducted to investigate the knowledge and experience of pathologists in Poland toward AI/ML in medical diagnosis. Our results showed that a higher proportion of the pathologists (≈58%) in this study did not use AI/ML, which influenced their knowledge about its usefulness in fastening the clinical diagnosis and regulation and policies controlling the implementation of AI/ML in the medical field. Additionally, there was a significant difference between pathologists who had previous experience with AI/ML in medical diagnosis and those who had not, pathologists with previous experience selected that physicians would be liable for legal problems caused by AI (83.33%). In comparison, the pathologists who had never had experience in AI/ML before selected the AI-developing company (55.56%). This result was significant compared to pathologists with previous experience with AI/ML. Trust and efficacy of AI/ML models in cancer diagnosis were positively associated with AI/ML in previous users. Therefore, pathologists highlighted the importance of AI/ML in the future as assistance tools without replacing their positions.

Comparing our results with other studies, a global study by Sarwar et al. (N = 487 pathologists) showed that 75% of participants were interested in AI as a diagnostic tool to improve the quality and efficacy of daily pathology work. At the same time, almost 20% were concerned that AI/ML tools could soon replace human positions. Moreover, 48.3% of their survey participants believed the final decision of the diagnosis must be for the human physician, and 25.3% thought the decision should be shared between physicians and the AI/ML model [11]. Another global study conducted on 669 physicians [24] showed that only 6% of the participants were familiar with AI applications in the medical field. Meanwhile, 83.4% agreed about the importance of AI/ML models in medical diagnosis and how it will save money and time for physicians. However, 29.3% expressed their concerns about the ability of the AI algorithm’s accuracy in complex and challenging cases. At the same time, 44% thought the AI diagnosis should be considered in the final decision stage over human physicians, while 35.4% expected that AI models would replace their job soon.

A survey of 116 pathologists by Meyer et al. found that 87% of the pathologists trust the AI diagnostic ability for cancerous tumors, and 92% of them agreed that AI algorithms play an essential role in their daily basic work, aiding and helping them make the most accurate decision [7]. In contrast, a cross-sectional study involving 250 Saudi healthcare providers showed that 75% believe AI abilities in medical diagnosis are superior to human experience, while 78% fear that AI could take over their jobs. The study also found that Saudi Arabia’s health sector has a potential market that could attract developers and researchers in AI/ML applications in the medical field [25]. Similarly, over 632 physicians reported concerns about the divestment of healthcare companies and medical liability implications despite the importance of AI/ML in the medical sector [26].

Finally, it is worth noting that the revolution of AI applications in the medical field has driven many authorized organizations worldwide to announce regulations that can ensure the safety, ethical, and legal procedures of AI applications in the medical field. The US Food and Drug Administration (FDA) regulates the use of AI in the medical field through its Center for Devices and Radiological Health (CDRH). The FDA’s approach to regulating AI in healthcare is risk-based, meaning that the level of oversight is commensurate with the product’s potential risks. For AI-based medical devices, the FDA has issued guidance on premarket submission requirements which outline the information that manufacturers must provide to the agency to receive clearance or approval for their products. These requirements include information on the device’s performance, design, intended use, and data on the device’s safety and effectiveness [27]. Moreover, the National Health Service (NHS) in the United Kingdom regulates the use of AI in the medical field through the National Institute for Health and Care Excellence (NICE) [28]. NICE is an independent organization that provides guidance and advice on the use of technology. NICE’s role in regulating AI in healthcare includes evaluating the safety, efficacy, and cost-effectiveness of AI-based products and issuing guidance and recommendations on their use. It is also worth mentioning that the legal responsibilities associated with integrating AI/ML in medical diagnosis have consequences for both pathologists and AI/ML-developing companies. Pathologists can face liability if they rely on AI systems for diagnosis without properly validating their accuracy and reliability. Pathologists may be held accountable if an incorrect diagnosis leads to patient harm. They also have ethical obligations to ensure that the AI systems adhere to professional and accepted standards. AI-developing companies can be held liable for any harm caused by their products if the AI system provides inaccurate diagnoses or treatment recommendations. Regulatory compliance is crucial, and non-compliance can lead to legal penalties and restrictions on product distribution. Pathologists and AI-developing companies must understand evolving legal standards to ensure patient safety and mitigate risks [29].

Finally, it is essential to highlight the strength and limitations of this study. This is the first cross-sectional study involving Polish pathologists’ perception of AI/ML. The data yielded from this study can be considered high-quality data since all pathologists were active and licensed. Still, our research has limitations. First, our sample size was approximately 15% to 22% of the active pathologists in Poland, which might be considered a low percentage; however, research suggests that 60 [30] or 40 [31] interviews can be enough for data saturation and variability. Nevertheless, a more extensive study may be concluded to shed light on other demographic factors in this group. Moreover, the responses gathered were exclusively from the pathologists and did not necessarily reflect the opinion of other healthcare providers. Hence, it should not be generalized. Additionally, some wording of our questionnaire may polarize the results differently. The average age of pathologists was below 40 years. Older pathologists with more experience might have different opinions. However, several factors influenced their participation, such as general interest, availability, and access to email during our study.

## 5. Conclusions

Our study results showed that pathologists in Poland are not fully aware of AI/ML applications in medical diagnosis due to their lack of experience using them. Additionally, the legal liability of using AI/ML in the medical field was not clear enough, especially since they expect a promising future of AI/ML applications as an assisting tool for the medical diagnosis process. Hence, they require more training and awareness to be up to date with AI/ML legal issues, guidelines, protocols, and applications in medical practice. Further research is required on a larger group of pathologists to develop medical education modules to fit their learning needs.

## Figures and Tables

**Figure 1 jpm-13-00962-f001:**
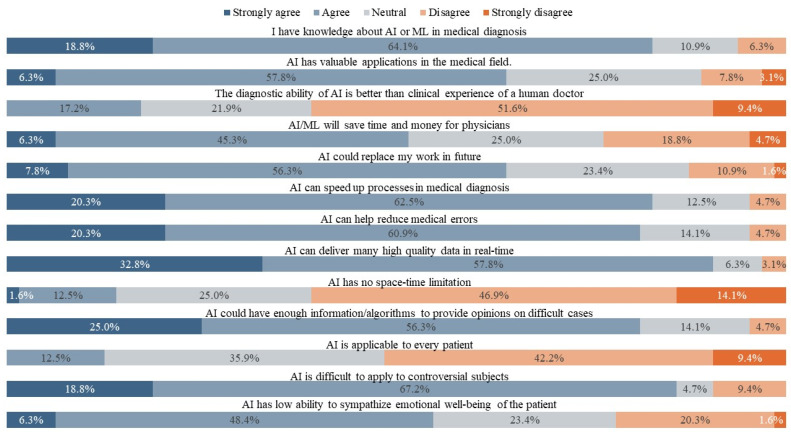
Reported frequencies of the survey questions related to the use of AI/ML.

**Figure 2 jpm-13-00962-f002:**
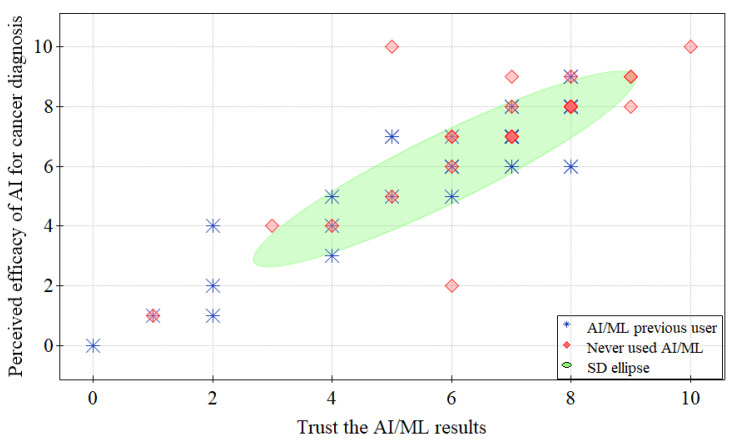
The relationship between trust in AI/ML results and its efficacy in cancer diagnosis.

**Figure 3 jpm-13-00962-f003:**
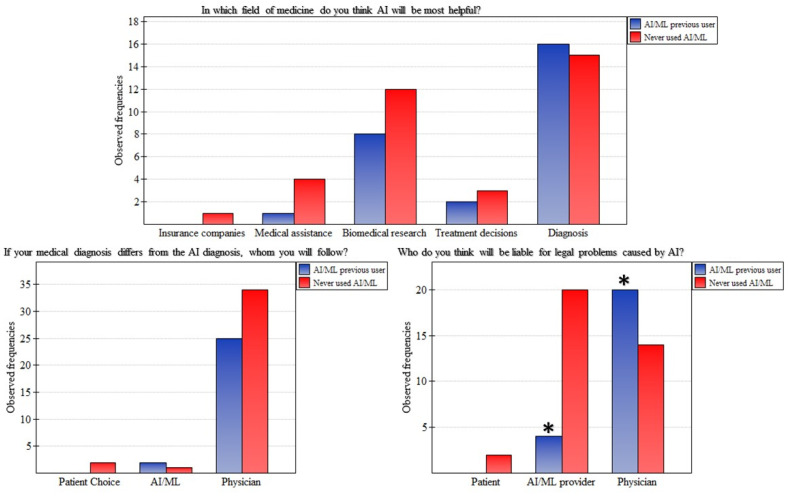
Concerns related to the use of AI/ML among pathologists. The star indicates significant differences in reported frequencies.

**Figure 4 jpm-13-00962-f004:**
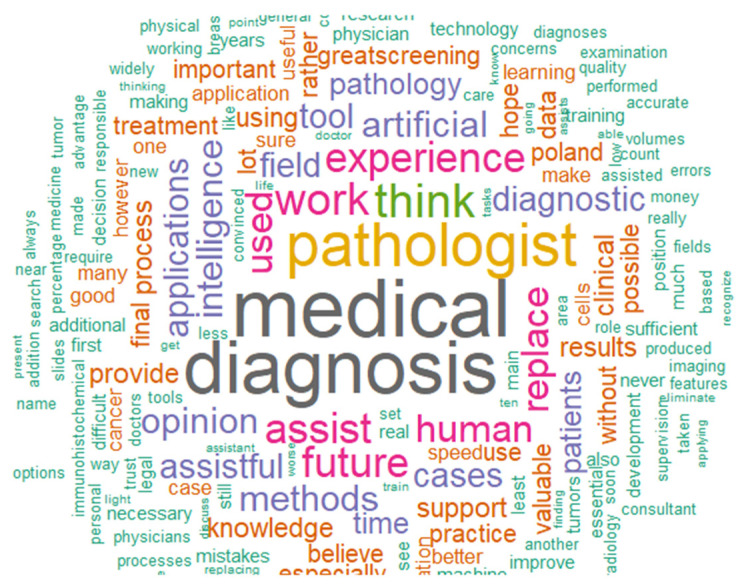
Text mining results of pathologists’ future expectations of using AI/ML in the clinical field.

**Table 1 jpm-13-00962-t001:** Survey validation using Cronbach’s standardized alpha.

Question No.	Question	Alpha
**Q1**	I have knowledge about AI or ML in medical diagnosis	0.778
**Q2**	AI has valuable applications in the medical field.	0.752
**Q3**	The diagnostic ability of AI is better than the clinical experience of a human doctor	0.763
**Q4**	AI/ML approaches will save time and money for physicians	0.735
**Q5**	AI could replace my work in the future	0.781
**Q6**	AI can speed up processes in medical diagnosis	0.754
**Q7**	AI can help reduce medical errors	0.745
**Q8**	AI can deliver much high-quality data in real-time	0.756
**Q9**	AI has no space–time limitation	0.764
**Q10**	AI could have enough information/algorithms to provide opinions on difficult cases	0.754
**Q11**	AI is applicable to every patient	0.781
**Q12**	AI is challenging to apply to controversial subjects	0.811
**Q13**	AI has a low ability to sympathize with the emotional well-being of the patient	0.814
	**Standardized alpha**	**0.793**

**Table 2 jpm-13-00962-t002:** Demographic data of the participating pathologists.

Age *	37 (33–41.5)	
Years of medical practice experience *	10 (6.75–15.25)	
Sex **	Male	25 (36.77)
	Female	36 (52.94)
	Prefer not to say	7 (10.29)
Did you use any AI/ML models before? *^,^**^,^***	Yes	27 (42.19)
	No	37 (57.81)
How much do you trust the AI/ML results in a scale-out of 10? *	7 (5.75–8)	
How do you evaluate the AI/ML in diagnosing cancer cells on a scale-out of 10 *	7 (6–8)	

* Reported as median (IQR), ** Reported as N (%). *** Missing four responses.

**Table 3 jpm-13-00962-t003:** Differences between agreement frequencies among pathologists with and without previous experience in using AI/ML with Bonferroni correction.

Question No.	Question	Never Used AI/ML			AI/ML Previous User			Mann–Whitney U Test		Chi-Squared Test	
		Median (IQR) *	No. (%) Disagree **	No. (%) Agree ***	Median (IQR) *	No. (%) Disagree **	No. (%) Agree ***	Z	*p*	χ^2^	*p*
Q1	I have knowledge about AI or ML in medical diagnosis	3 (2-4)	14 (58.33)	10 (41.67)	4 (4-4)	0 (0.00)	25 (100.00)	−5.284	<0.001 ****	20.417	<0.001
Q2	AI has valuable applications in the medical field.	4 (4-4)	4 (11.76)	30 (88.24)	4 (4-4.5)	2 (7.41)	25 (92.59)	−1.588	0.112	0.322	0.570
Q3	The diagnostic ability of AI is better than the clinical experience of a human doctor	2 (2-3)	21 (84.00)	4 (16.00)	3 (2-3)	12 (75.00)	4 (25.00)	−1.206	0.228	0.503	0.478
Q4	AI/ML approaches will save time and money for physicians	4 (4-4)	2 (6.67)	28 (93.33)	4 (4-5)	1 (4.00)	24 (96.00)	−2.403	0.016	0.188	0.665
Q5	AI could replace my work in the future	2 (2-3)	22 (78.57)	6 (21.43)	2 (2-3)	17 (85.00)	3 (15.00)	−0.123	0.902	0.316	0.574
Q6	AI can speed up processes in medical diagnosis	4 (4-4)	1 (3.03)	32 (96.97)	4 (4-5)	1 (3.70)	26 (96.30)	−2.337	0.019	0.021	0.885
Q7	AI can help reduce medical errors	4 (4-4)	2 (6.06)	31 (93.94)	4 (4-5)	1 (4.55)	21 (95.45)	−1.097	0.272	0.059	0.808
Q8	AI can deliver much high-quality data in real-time	4 (4-4)	2 (6.45)	29 (93.55)	4 (4-5)	1 (4.00)	24 (96.00)	−1.692	0.091	0.164	0.685
Q9	AI has no space–time limitation	4 (3-4)	5 (18.52)	22 (81.48)	4 (3-4)	3 (13.64)	19 (86.36)	−1.271	0.204	0.212	0.646
Q10	AI could have enough information/algorithms to provide opinions on difficult cases	3 (3-4)	9 (36.00)	16 (64.00)	4 (3-4)	6 (26.09)	17 (73.91)	−1.409	0.159	0.548	0.459
Q11	AI is applicable to every patient	2 (2-3)	21 (84.00)	4 (16.00)	2 (2-3.5)	18 (72.00)	7 (28.00)	−0.052	0.959	1.049	0.306
Q12	AI is challenging to apply to controversial subjects	4 (3-4)	3 (10.34)	26 (89.66)	4 (3-4)	4 (21.05)	15 (78.95)	−1.689	0.091	1.057	0.304
Q13	AI has a low ability to sympathize with the emotional well-being of the patient	4 (4-4)	2 (5.71)	33 (94.29)	4 (3.5-4)	2 (9.09)	20 (90.91)	−1.799	0.072	0.236	0.627

* Participant perceptions were measured using the following scale: 1 = Strongly Disagree; 2 = Disagree; 3 = Neither Agree nor Disagree; 4 = Agree; 5 = Strongly Agree. ** Percentage in disagreement was calculated using those who responded, “Strongly disagree” or “Disagree”. *** Percentage in agreement was calculated using those who responded, “Strongly agree” or “Agree”. **** Significant at *p* < 0.00384.

**Table 4 jpm-13-00962-t004:** Univariate and backward stepwise multi-regression analysis for predictors of being a previous user of AI/ML.

Question No.	Question	b Coeff.	b Error	Wald Stat.	*p*	Odds Ratio	−95% CI	+95% CI
	Univariate predictors							
**Q1**	I have knowledge about AI or ML in medical diagnosis	2.842	0.769	13.655	<0.001	17.158	3.799	77.488
**Q2**	AI has valuable applications in the medical field.	0.488	0.361	1.829	0.176	1.628	0.803	3.301
**Q3**	The diagnostic ability of AI is better than the clinical experience of a human doctor	0.387	0.314	1.520	0.218	1.472	0.796	2.723
**Q4**	AI/ML approaches will save time and money for physicians	0.825	0.393	4.408	0.036	2.281	1.056	4.926
**Q5**	AI could replace my work in the future	−0.072	0.273	0.069	0.792	0.931	0.545	1.589
**Q6**	AI can speed up processes in medical diagnosis	0.848	0.434	3.819	0.051	2.336	0.998	5.468
**Q7**	AI can help reduce medical errors	0.355	0.363	0.959	0.328	1.426	0.701	2.903
**Q8**	AI can deliver much high-quality data in real-time	0.593	0.390	2.312	0.128	1.810	0.843	3.886
**Q9**	AI has no space–time limitation	0.316	0.313	1.014	0.314	1.371	0.742	2.535
**Q10**	AI could have enough information/algorithms to provide opinions on difficult cases	0.334	0.267	1.572	0.210	1.397	0.828	2.355
**Q11**	AI is applicable to every patient	0.110	0.286	0.148	0.700	1.116	0.637	1.957
**Q12**	AI is challenging to apply to controversial subjects	−0.576	0.323	3.186	0.074	0.562	0.299	1.058
**Q13**	AI has a low ability to sympathize with the emotional well-being of the patient	−0.574	0.365	2.484	0.115	0.563	0.276	1.150
	Backward stepwise multi-regression analysis							
**Q1**	I have knowledge about AI or ML in medical diagnosis	2.885	0.823	12.297	<0.001	17.903	3.570	89.787
**Q2**	AI can speed up processes in medical diagnosis	1.541	0.762	4.090	0.043	4.668	1.049	20.777

## Data Availability

Data can be obtained from the corresponding author upon reasonable request.

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
