# Peer review of "Perception of Pathologists in Poland of Artificial Intelligence and Machine Learning in Medical Diagnosis—A Cross-Sectional Study"

_jpm, 2023, doi:10.3390/jpm13060962_

Round 1

Reviewer 1 Report

Pathology is a medical subspecialty based on imaging analysis and diagnosis, which will be benefit significantly from the current revolutionary applications of AI technology.  The opinions of pathologist towards AI technology will impact on the implementation and application of the AI technology in pathology practice, especially after the recent introduction of Chat-GPT, a powerful AI tool already affecting many aspects of human life. Your research is based on a cross-sectional survey of 68 Polish pathologists’ opinions to towards AI, although the number is small compared to other studies, the result is still valid.  I would like to make several comments to improve the quality of the manuscript to meet the standard for the publication.   

1.       The introduction part.  The introduction is quite comprehensive, touching on every aspects of AI and pathology.  Yet it is not as structured and concise as it should be.  It will be benefited from a re-writing, breaking down the part into several paragraphs.

2.       The discussion part.

(1). there is no need to have a sentence of number 5. Strengths and limitations. Just have the paragraph discussing about limitations of the study will be fine.

(2). the fourth paragraph of the discussion part writes about legal issue of AI.  The regulatory part of government could be shortened. Discussion on consequence of legal responsibility of AI diagnosis, either is pathologist or AI Company should be added.

Although the manuscript is a written with good English, there are still many obvious grammar mistakes and unconventional use of the language for scientific writing.  For examples:

1.       Line 43: such as [2}; is not good English and needs to be fixed;

2.       Line 52: used in scientific,  should be used in scientific research;

3.       Line 298: take off, should be take over

4.       The paragraph 2 and 3 in the discussion part: the writing of the comparison of the numbers of the current study with those of the previous studies is unconventional and needs to be fixed.

I would recommend using grammar software or Chat-GPT to check and fix the English mistakes in the introduction and discussion parts of the manuscript.

Reviewer 2 Report

The concept of the article is relevant and interesting. However, only 15% of the active pathologists in Poland were considered. The study is underpowered. 

Also there are statistical points which need clarification. These have been embedded in the article.

References are not as per the Vancouver style. 

Minor corrections in English. Have been highlighted
